# Investigation of the Surface Treatment Process of AISI 304 Stainless Steel by Centrifugal Disc Finishing with the Use of an Active Workpiece Holder

**DOI:** 10.3390/ma15196762

**Published:** 2022-09-29

**Authors:** Mateusz Juniewicz, Jarosław Plichta, Marzena Sutowska, Czesław Łukianowicz, Krzysztof Kukiełka, Wojciech Zawadka

**Affiliations:** Department of Production Engineering, Faculty of Mechanical Engineering, Koszalin University of Technology, Racławicka 15-17, 75-620 Koszalin, Poland

**Keywords:** centrifugal disc finishing, active holder, stainless steel, surface topography assessment

## Abstract

This article presents the results of experimental studies of the centrifugal disc finishing (CDF) process of 304 steel elements with the use of an active workpieces holder, that allows workpieces for additional rotational and oscillation movements. The main aim of the research was to evaluate the mechanism of formation of the surface texture and to assess the intensity and effectiveness of the machining process. It is shown that additional movements of the workpiece significantly affect the formation of the machining traces generated by the elementary phenomena of micro-cutting, scratching, grooving, etc. As a result, these combined and complex interactions lead to the formation of the surface topography of the workpieces. Based on the research results, it can be concluded that the use of an active workpiece holder in the CDF process allows changes in the intensity of the machining process. Moreover, the active holder allows modification of the surface smoothing process. The intensity of the treatment process depends primarily on the location of the workpiece holder in the appropriate energy area of the work charge. On the other hand, the efficiency of the workpiece surface smoothing depends on the parameters of the oscillation and rotational movements of the workpiece mounted in the active holder. The presented research results show that the use of an active holder, enabling rotation and oscillation of the workpiece, may lead to a more effective use of smoothing processes in CDF machines. The analysis of the results shows that the values of the *Sdr* and *Sa* parameters are more strongly dependent on the vibration frequency and increase with its increasing frequency. This is undoubtedly the result of the concentration of smoothing marks on the smoothed surface. However, with regard to the rotational speed of the object, this relationship is non-monotonic, and its greatest influence occurs at its intermediate values. It follows that this activity does not have a significant impact on the generation of the number of smoothing marks and the degree of their concentration. The research methodology proposed in the work allows the initial determination of the dependence of the results of the CDF process on the machining parameters, including the parameters of the active holder. This methodology can also be used for machining materials other than AISI 304 steel.

## 1. Introduction

Processes for smoothing the surface of small objects of various shapes made of different metal materials are effectively carried out by centrifugal disc finishing (CDF) machines [1,2,3]. The smoothing process in CFD is based on the interaction of abrasive or polishing pieces placed in the working chamber with the surfaces of the workpieces, most often with a solution of supporting fluid, during forced rotation (Figure 1).

In the smoothing process, there are a number of simultaneous elementary phenomena, which include:micro-smoothing with abrasive grains from lumps of abrasive chips,mechanical abrasion of the unevenness of the smoothed surfaces due to the unevenness on the surface of the chips,plastic ridging by the blades of the chips with a small caving,lapping of the surface of the workpieces by abrasive grains and micro-grains, which are products of wear,smoothing surface unevenness by plastic deformation,chemical reactions and anodic processes occurring on the surfaces of workpieces in the presence of active chemical solutions and abrasive or polishing pieces.

Depending on the interaction between the contact surface with a specific instantaneous orientation and the value of the contact force and velocity, different interaction mechanisms will be induced, i.e., plastic deformation, micro-cutting, grooving or scratching. In the case of the normal direction of action of the chips on the treated surface, large local plastic deformations can be observed, which are result of the formation of microcraters. On the other hand, the tangential movement of the chips in relation to this surface, are generating the micro-cutting and scratching processes, because the speed of this movement is relatively high compared to the speed in the normal direction. In the intermediate states of orientation of the treated surfaces, the cumulative effect of the abrasive blocks has a different character, which results in different characteristics of the mechanisms of plastic deformation and micro-cutting and scratching.

These processes belong to the group of finishing abrasive surface treatments with the use of mass finishing media, such as abrasive small particles for grinding and polishing chips [4,5,6,7]. The treatment consists of removing thin layers of material from the entire surface of the workpieces, shaped by the previous treatment. The aim of the process is to obtain the surface texture with desired performance properties, which concern for example reflectivity, adhesiveness, aesthetics, hygiene requirements, etc.

In research on the finishing abrasive surface treatments [2,6,7,8,9], including CDF, several interrelated basic directions of development can be distinguished. These works are carried out by means of theoretical considerations and analyses, modelling studies, e.g., with the simulation use of the discrete element method (DEM), and by means of experimental studies. The first line of research concerns the energy analysis of a given abrasive process. Generally, the main research purpose is to increase the efficiency and reduce the energy consumption of machining and its duration. The works in this field analyse the metal removal rate (MMR), weight loss per unit time, energy conversion, etc. A model media movement for CDF combined with MMR was analysed in previous work [6].

The second direction of research is aimed at obtaining the required surface texture of the machined surface in the finishing processes. Unfortunately, this type of research is the least developed, as mass-finishing processes are generally poorly controllable and difficult to optimize. The aim of the research is to conduct a given treatment in such a way that its effects in the form of the surface texture are predictable [10,11], consistent with expectations, and repeatable [12,13,14]. In [10], the authors proposed a model combining vibration finishing process parameters and material removal mechanisms. This model can be used to predict the roughness of the material surface for given parameters. The investigations of the contact of abrasive particles with the workpieces were carried out using the DEM simulation. A wide review of previous research on mass finishing processes, including numerical modelling, to study the effect of various input process parameters involved in mass finishing was presented in this work [15].

The third important research area of technologies for surface finishing is the study of the kinematics and dynamics of a given machining process [12,16]. These studies aim to evaluate velocities and accelerations, forces occurring during machining, as well as other parameters of the machining process [17,18]. The process kinematics and the media distribution characteristics in centrifugal finishing and spindle finishing were analysed based on theoretical analysis and DEM simulation in the work [17]. This allows rational use of the existing technological devices, reduction in their wear, and effective selection and conservation of abrasives, as well as to some extent supervision of the hard to control machining process.

Usually, the treatment process in the rotary disc finishing machine is carried out with free orientation of the workpieces in the stream of rotating work charge composed of forming chips. The movement of the work charge and workpieces is caused by the kinetic energy of the rotating disc located at the bottom of the conical work chamber of the CDF machine. The machining process takes quite a long time. It is a poorly controllable and difficult to optimize. The instantaneous energy values of the work charge are locally variable [19]. The energy distribution varies in the longitudinal and transverse directions of spinning mass and it changes during work. These changes are caused by the complex spinning motion of the mass finishing media in the rotating stream in the chamber of the finishing machine, as well as the mutual interactions of the workpieces and the mass finishing chips with the workpieces.

A good modification of the kinematics of the finishing process, leading to the improvement of machining efficiency, is placing workpieces in holders, and their appropriate positioning in the abrasive stream [20,21,22]. Due to functions in the machining process, the stationary and active holders are distinguished. Stationary holders allow placement of the workpieces in established positions. Active holders enable additional movement of the workpieces, which ensures an even treatment of all surfaces [21,22,23]. On the other hand, active holders allow extra moves for the workpieces, including an auxiliary rotary motion, which increases the efficiency of machining [21,22].

Research conducted at the Koszalin University of Technology indicates that in the case of the CDF machining process of objects in active holders, allow for greater possibilities of controlling the intensity of machining and the surface texture of the workpieces [24,25]. For this purpose, active holders are used, which let the workpieces perform other movements. These are positioning movements and various types of working movements, as well as their mutual joints. It allows an influence on the surface texture formation process of workpieces, due to changing the share of elementary machining phenomena in this process in the form of friction, micro-cutting, cratering, scratching, etc. In addition, it is possible to change the surface texture by orienting the machining traces, their compaction, and deepening. The aim of the experimental research was to analyze the mechanism of shaping of the surface texture in the CFD process in the container, with the use of additional movements (vibration) of the workpieces mounted in the holder. The purpose of this research is to assess the intensity and efficiency of this process. Such a scope of experimental research enables the analysis of the influence of the surface smoothing process in the adopted technical variants and allows the determination of the influence of the method of abrasive shapes on the shaping of the surface topography of the treated surface. This paper presents the results obtained during the tests of the processing of samples made of AISI 304 steel in a CDF machine with the use of an active holder.

## 2. Materials and Methods

### 2.1. Purpose of the Research

The aim of the research was to analyze the formation of the surface texture during the CDF process with the use of additional movements of the workpiece mounted in the active holder. The research was carried out in two stages. The aim of the first step was to check the surface smoothing efficiency for three different treatment methods. The effectiveness was assessed based on changes in the 3D parameters of the surface roughness and surface topography. In the second stage, the aim of the research was to assess the influence of the parameters of the vibratory and rotary motions of the machined samples and the disc rotary motion of the CDF machine on 3D parameters of the surface roughness.

### 2.2. Research Stand and Methodolgy of Experimental Research

The research stand consists of two basic units: the CDF machine EC6, made by AVALON [26], and active holder for a workpiece, that is mounted on the working chamber of this finishing machine. A view of the EC6 machine and a schematic diagram of a cross-section of its working chamber are shown in Figure 1.

The workpiece holder is protected by a patent application [27]. Figure 2a shows the view of the test stand, and Figure 2b shows a schema of the active workpiece holder. This schema shows the basic types of positioning displacements and working movements.

The applied holder has the possibility of radial positioning of the workpiece in relation to the rotor axis of the disc finishing machine, by setting displacements along the X direction. The depth of the object immersion in the working charge is set by moving the holder arm vertically in the Z direction. During machining, the workpiece can be rotated around the chuck axis at a rotational speed in the range *n_h_* = 0–100 rpm, to ensure even machining of its entire surface. Moreover, the holder, together with the workpiece, can be subjected to harmonic vibrations in the vertical direction with adjustable amplitude *A* and frequency *f* in the following ranges: *A* = 0–1 mm, *f* = 0–100 Hz. For this purpose, the active holder is equipped with a small electric motor, a rotary gearbox and a controlled vibration generator.

Figure 3 shows the basic parts and setup of the CAD model of the active holder that was used in the research. The two main kinematic units of the active holder i.e., the rotary and the vibrational motion of the active holder, are shown in Figure 4.

The research investigations were carried out on nine flat samples, 25 × 25 × 2 mm^3^ in size, made of AISI 304 stainless steel. The chemical composition is shown in Table 1.

In the sample smoothing process, finishing media chips with a resin binder, marked by the manufacturer as 02PP10 (triangular pyramids 10 × 10 mm^2^), were used. The view of the finishing media in the working chamber of the EC6 smoothing machine and an enlarged image of abrasive chips are shown in Figure 5. Each sample, after mounting in the active holder, was placed in the area with the highest energy of the working charge, determined in accordance with the methodology described in the works [24,25].

The research plan was divided into the following two stages:

The research on the influence of the type of workpiece movement on the surface texture in relation to machining in a stationary holder.The research on the influence of the integrated rotation and vibration movements of the workpiece on the shaping of the surface texture.

In the first stage of the research, three variants of the samples’ machining were used: CDF of the stationary samples, CDF with rotational movement of the samples and CDF with rotation and vibration of the samples. The CDF process was carried out at the rotational speed of the rotor *n* = 300 rpm for 0–45 min, stopping the process every 15 min and measuring the surface texture. The surface topography and selected 3D surface roughness parameters of the samples were assessed. Changes in these parameters were the basis for the evaluation of the smoothing efficiency in three machining variants.

The TalySurf CLI 2000 from Taylor–Hobson optical profilometer with TalyMap software was used for the surface texture measurements. Using this measurement system, the surface roughness parameters were determined in accordance with the standard [28], which are described in more detail in [29].

The smoothing efficiency of the sample surfaces shaped in the tested variants of the machining process were calculated from the following relationship:*ΔSx* = [*Sx*(0) − *Sx*(*t*)] · [*Sx*(0)]^−1^ · 100%,(1)
where: *ΔSx*—the relative difference values of the selected parameter of the surface roughness; *Sx*(0)—initial value of the selected parameter of the surface roughness; *Sx*(*t*)—final value of the selected parameter of the surface roughness after machining time *t*.

Moreover, the machined surfaces were also observed using a scanning electron microscope (SEM) and assessed by light scattering measurements, in accordance with the methodology provided in [30].

The second stage of the research was to determine the impact of the integrated rotary–vibration motion of the active holder on the sample surface texture. To determine this relationship the three-level experimental plan was used [31,32,33]. The variable parameters were the rotational speeds *n* of the CDF machine disc and the holder *n_h_*, as well as the vibration frequency *f* of the active holder. The results were elaborated statistically in Statistica 13.1 software. The research methodology and the samples were similar to the first stage of the research.

After choosing the factors influencing the research object, the ranges of the variability of the input factors were determined. Accepted ranges of the input factors for the research are:
-For the first stage of research:
(1)machining time: ***t* = 15, 30 and 45**, min,
-For the second stage of research:
(2)rotational speed of the disc: ***n* = 312, 416 and 520**, rpm(3)vibration frequency of the active holder: ***f* = 35, 65 and 95**, Hz.

The values of input factors for each measurement point for the first stage are presented in Table 2 and for the second stage are in Table 3.

## 3. Results and Discussion

The research results compose a comprehensive data set. Only synthetic selected results of the first and second stage studies are given below. In Table 2 are shown the results obtained in the first stage of the research. Based on these results, graphs showing the changes in the surface texture parameters during processing were developed. The exemplary graphs of changes in the *Sdr*, *Sdq* and *Ssc* parameters as functions of machining time are shown below in Figure 6, Figure 7 and Figure 8. These parameters showed the high sensitivity of indicators of the surface smoothing efficiency. The parameter changes shown in Figure 6, Figure 7 and Figure 8 follow a similar course, but the *Sdr* parameter is the most sensitive.

The analysis of changes in the values of the *Ssc* parameter as a function of the machining time, shown in Figure 8, shows an increase in the average value of the rounding radii of the surface summits. The radii o−f the summit’s rounding of the surface texture are assessed based on the inverse of the *Ssc* parameter. The increase in the radii of the peaks indicates the progressive process of smoothing the peaks of the unevenness. This increase is diversified according to the tested machining variants.

Figure 9 and Figure 10 summarize selected data obtained from the TalySurf CLI 2000 profilometer and the TalyMap software, which allowed for the preparation of Table 2, as well as Figure 6, Figure 7 and Figure 8.

The images of the sample surfaces, obtained by means of SEM, after smoothing during the first stage of the research, are shown in Figure 11. Based on SEM images of the treated surfaces in such a research plan, it can be concluded that both the implementation of smoothing without additional workpiece movement and the active workpiece holder significantly influence the process of shaping the geometric structure of the smoothed surfaces. This is confirmed by microscopic images of surfaces after 45 min of treatment in the accepted technological variants.

Thus, in the case of smoothing without the movement of the object (Figure 11a), some characteristic smoothing mechanisms can be distinguished on the shaped surface. Micro-cavities, which are the result of dynamic contacts of the tips of the abrasive blocks, have a dominant share here. On this basic structure, there are local traces of micro-cutting and grooving with few flashes as well as traces of scratching and reeling of the chips on the smoothed surface. The dynamics of these interactions is relatively high and results from the long contact time of the abrasive chip with the smoothed surface. In the case of smoothing with the rotation of the workpiece (Figure 11b), the time of temporary contacts of the abrasive pieces is significantly shortened and a local change in the direction of their operation takes place as a result of the mutually complex movement. As a result, there is a local increase in the number of smoothing marks and plastic deformations. The smoothed surface is characterized by a greater degree of development and a relatively higher roughness. On the other hand, in the case of smoothing with rotational and vibrating motion (Figure 11c), there is a cumulative interaction of these movements with the rotational (centrifugal) motion of the abrasive chips. The time of their temporary contacts is shortened, and the directions of its operation are changed, which leads to a relatively greater reduction in surface roughness.

During the research on various variants of the CDF process, an attempt was also made to assess the degree of surface smoothing using the light scattering method, described in [30]. This method consisted of illuminating the tested surface with a laser beam and examining the angular distribution of the scattered light intensity. The degree of surface smoothing was assessed based on the standard deviation *Sn* of this distribution. Figure 12 shows the results obtained with this method.

Table 3 presents the selected results of the second stage of the research. They show a large influence of the rotational speed *n* of the disc in the CDF machine and the vibration frequency *f* of the active workpiece holder on the surface texture parameters of the samples. On this basis, the 3D plots were developed, showing the dependence of the surface texture parameters on the rotational speed *n* of the disc and the vibration frequency *f* of the active holder. The exemplary three-dimensional plots of the dependence *Sa* and *Sdr* parameters on the *n* and *f* parameters is shown in Figure 13.

The results of the conducted research indicate that the effectiveness of the CDF process can be assessed based on the analysis of changes in the *Sdr*, *Sdq* and *Ssc* parameters. The most sensitive parameters are the *Sdr* and *Sdq* parameters. On the other hand, a decrease in the value of the *Ssc* parameter during machining indicates an increase in the value of the local radii of curvature of the surface roughness.

As a result of introducing additional movements of the workpiece in the active holder, especially rotating and oscillating movements, the frequency of interaction between the finishing chips and the machined surface has an important influence on the received surface after processing. This leads to an increase in the efficiency of the machining process, a reduction in the height of unevenness, an increase in the local radii of curvature and an increase in surface density of the machining marks. This is also confirmed by microscopic observations carried out with the use of SEM.

The analysis of the results in Table 3 showed that the formation of the surface texture in the CDF smoothing machine using the active workpieces holder can be partially controlled by the parameters of the machining process. It is observed that the smoothing effects caused by the rotation–vibration movement of the active holder depend on the vibration frequency *f* and the rotational speed *n* of the CDF machine disc. The influence of the rotation speeds *n_h_* of the holder is smaller.

Table 3 and Figure 13 show that the range of changes in the *Sa* parameter in the conducted experiments were within the range of 0.97–2.38 µm. The course of the dependence of the *Sa* parameter on the processing parameters *f* and *n* has a similar form to a like dependence determined for the *Sdr* parameter. However, the values of the *Sdr* parameter change in a greater range of variation 1.72–7.76%. This may indicate that the *Sdr* parameter is more sensitive to changes in the texture of the workpiece surface in the CDF process.

## 4. Conclusions

The conducted research allows several conclusions mainly concerning the use of the active holder in the CDF process. It was found that the rotation–vibration movement of the workpiece in the CDF smoothing machine, introduced by the active holder, significantly influences the method of the surface texture formation of the workpiece. The formation method of the surface texture in the CDF smoothing machine with the use of an active workpiece holder, which gives the rotary and vibration motion, is influenced to a lesser extent by rotational speed than by vibration frequency.

It was observed that the vibration frequency significantly influences the degree of development of the machined surface characterized by the *Sdr* parameter, the number of peaks of the roughness specified by the *SPc* parameter, and the value of the *Sa* parameter describing the arithmetic mean height of the surface roughness. The influence of rotations and vibrations of the workpiece on the degree of isotropy *I* of the machined surface is relatively small and non-monotonic. However, it is possible to indicate the ranges of values of these parameters for which surfaces with the highest level of isotropy are obtained.

The results of the smoothing process in the CFD machine of AISI304 steel elements, mounted in an active holder with rotational and rotary–vibration movement and in a stationary holder, presented in the first stage of the research, indicate significant differences in the mechanism of shaping the surface texture of the smoothed surfaces. These differences result from the diversified relative direction of movement of the abrasive chip with the smoothed surface, the time of their contact and the number of accumulated contacts. It influences the shaping of the smoothing traces originating from the elementary phenomena of micro-cutting, scratching, grooving, cratering, and the mechanism of shaping the surface structure. The best results in terms of shaping the geometric structure are achieved with the use of rotational and vibrational movements of the workpiece. This method of processing also provides a significantly higher processing intensity compared to the processing carried out with the rotational movement of the workpiece itself, in particular with processing in a free state. It is a synergistic effect of the influence of a complex system of mutual movements of the object and working chips, the multiplicity and dissimilarity of elementary phenomena occurring in the smoothing zone.

The method of shaping the surface texture of the surface in the CFD machine with the use of an active workpiece holder, which gives the cumulative rotary–vibration motion, a lesser influence is observed by rotational speeds than by the vibration frequency of the workpiece.

The formation of the surface texture of the workpiece surfaces in the CDF machine with an active object holder depends to a large extent on other working, rotating, and rotating–vibrating movements performed by the workpiece. The presented research results show that using an active holder, enabling rotations and oscillations of the object, may lead to more effective machining processes in CDF machines. The obtained results quantitatively characterize the intensity of changes in the surface texture parameters of workpieces smoothed with the use of an active holder, depending on the machining parameters. This knowledge can be used to design a machining process for AISI 304 steel.

The presented research results show that the use of an active holder, enabling rotation and oscillation of the workpiece, may lead to a more effective use of smoothing processes in rotary-cascade container smoothing machines. This knowledge can be used to design a smoothing process that allows specific values of parameters characterizing the surface texture of the surface of AISI 304 stainless steel to be obtained.

## 5. Patents

The Koszalin University of Technology submitted a patent application to the Patent Office of the Republic of Poland concerning the active holder [25].

## Figures and Tables

**Figure 1 materials-15-06762-f001:**
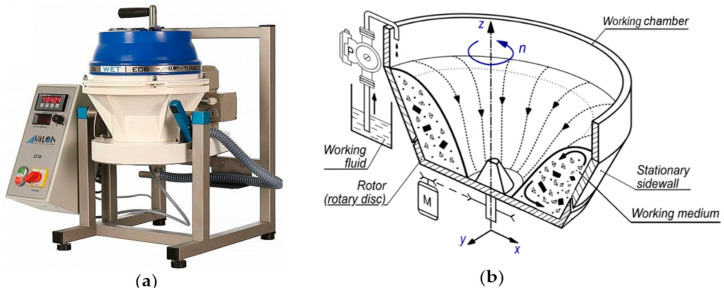
View of the EC6 smoothing machine (**a**) and a schematic diagram of its working chamber (**b**).

**Figure 2 materials-15-06762-f002:**
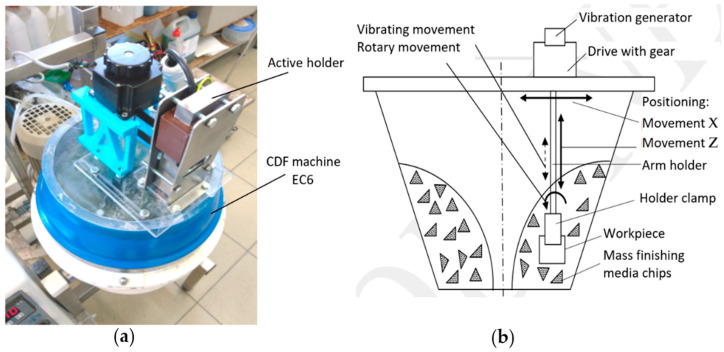
Test stand with an active workpiece holder: view of the test stand (**a**), scheme of the active workpiece holder mounted on the smoothing machine working chamber, showing the types and directions of the workpiece movement carried out by the holder (**b**).

**Figure 3 materials-15-06762-f003:**
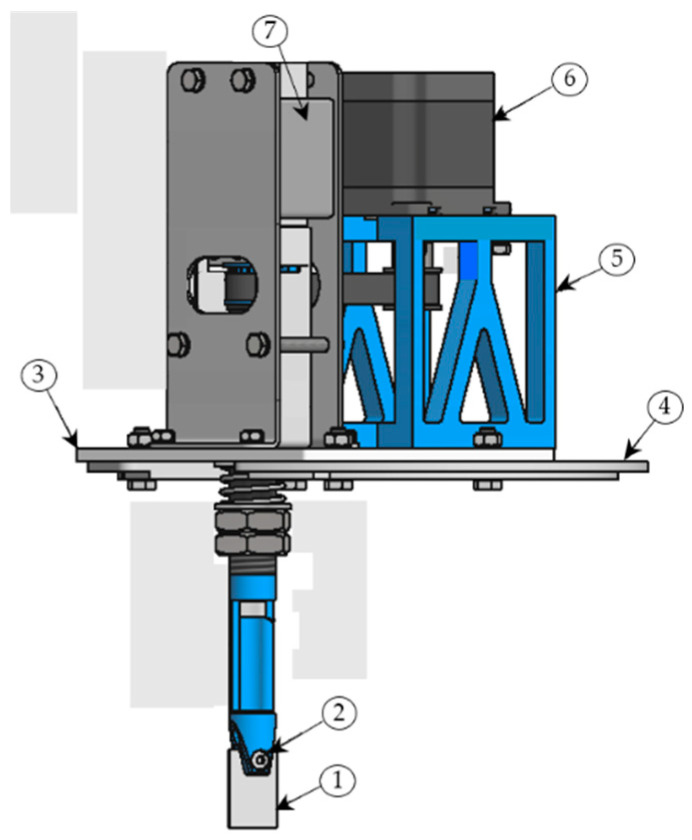
The CAD model of the active workpiece holder: 1—sample, 2—screw clamp, 3—sliding plate, 4—cover plate, 5—motor bracket, 6—stepper motor, 7—electromagnetic vibrator.

**Figure 4 materials-15-06762-f004:**
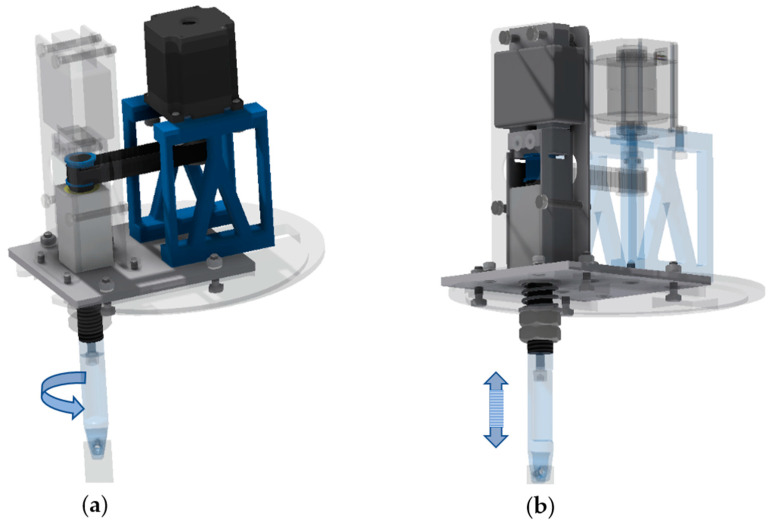
Two main kinematic units of the active holder: rotary (**a**) and vibration (**b**) motion of the active holder.

**Figure 5 materials-15-06762-f005:**
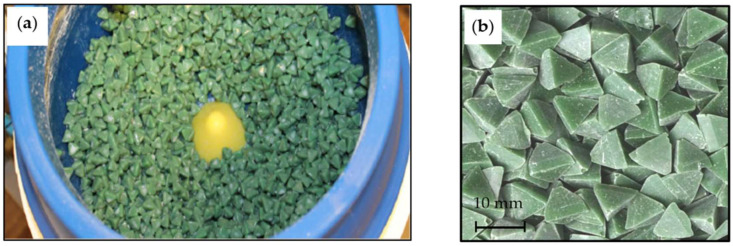
General view of the finishing media signed as 02PP10 in the working chamber of the EC6 smoothing machine (**a**) and an enlarged image of abrasive chips (**b**).

**Figure 6 materials-15-06762-f006:**
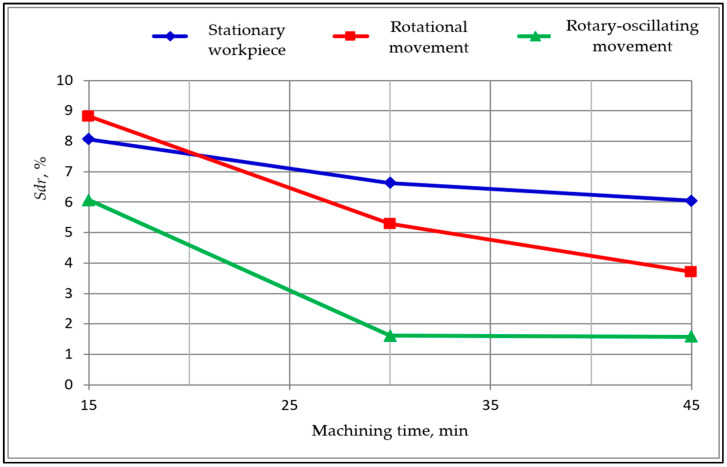
Changes of the developed interfacial area ratio *Sdr* as a function of machining time.

**Figure 7 materials-15-06762-f007:**
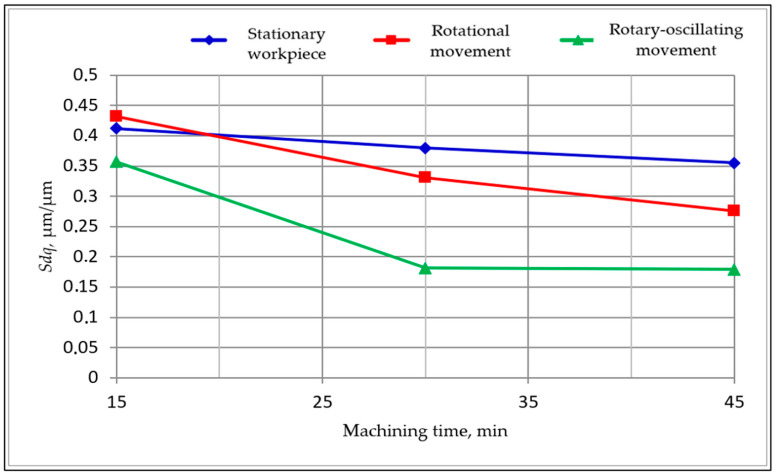
Changes of the root mean square gradient *Sdq* as a function of machining time.

**Figure 8 materials-15-06762-f008:**
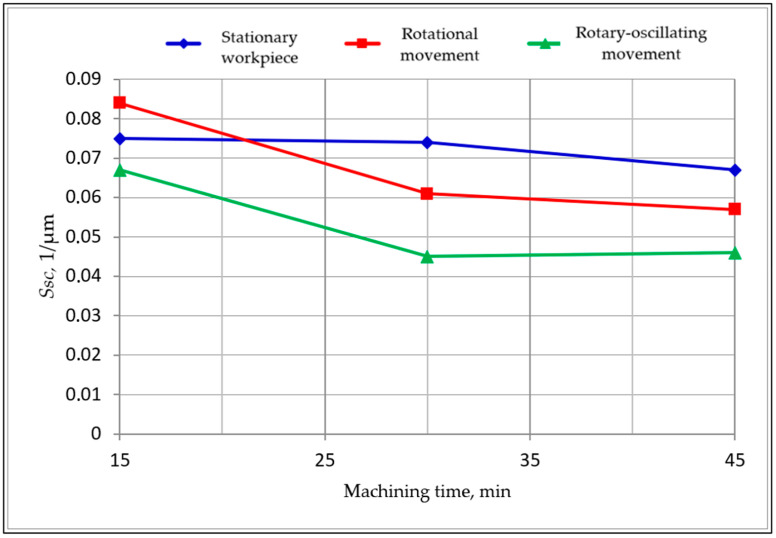
Changes of the arithmetic mean summit curvature *Ssc* as a function of machining time.

**Figure 9 materials-15-06762-f009:**
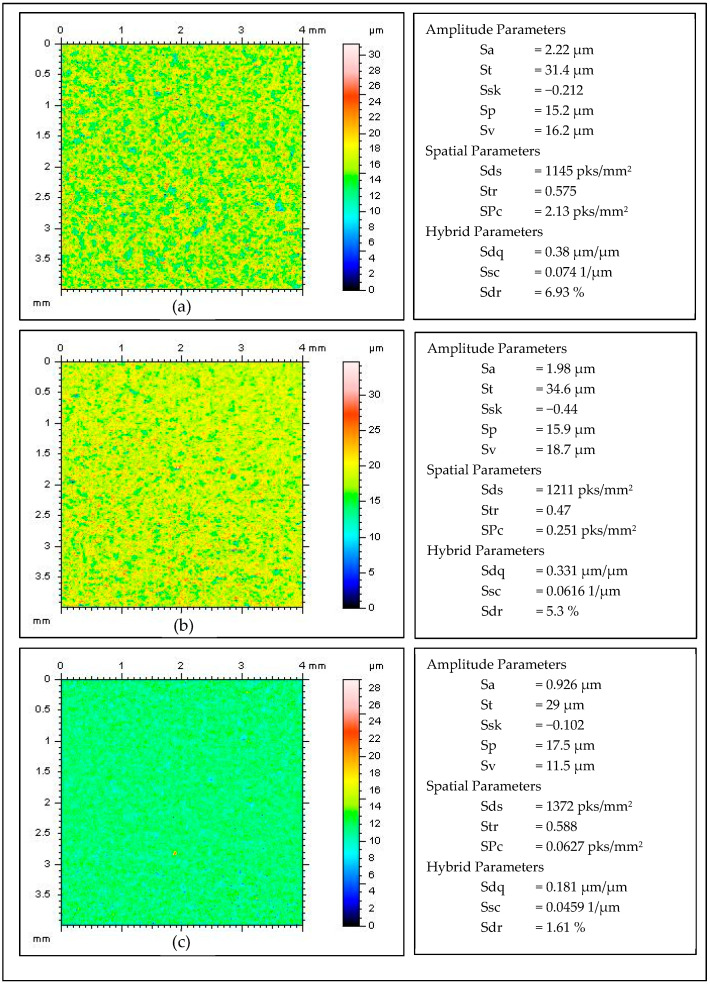
False color maps of the sample surface and the surface roughness values, obtained by TalySurf CLI 2000 for three samples after 30 min machining by CDF without movement (**a**) with rotational movement (**b**) and with rotation and oscillating movement (**c**).

**Figure 10 materials-15-06762-f010:**
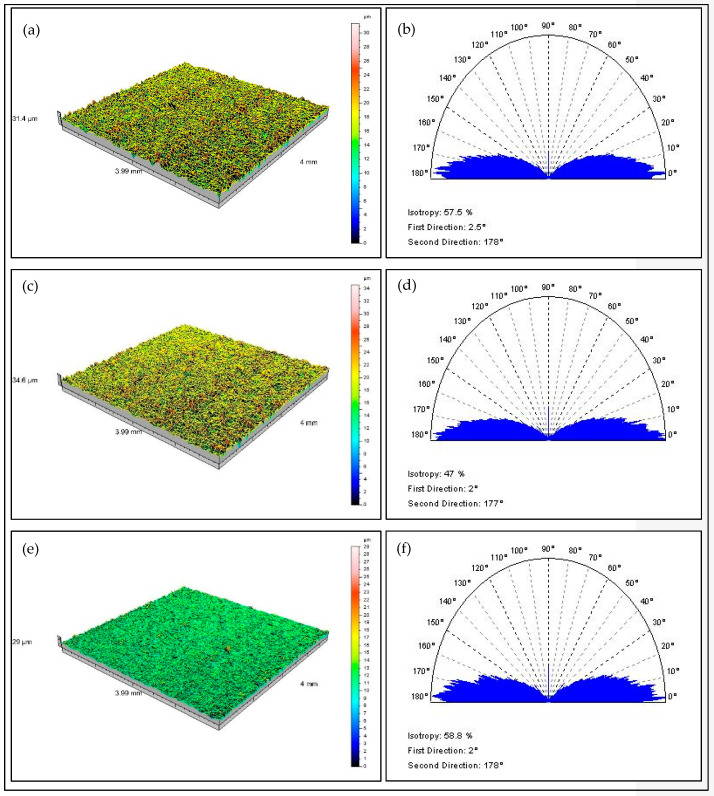
3D surface views and the orientation plots of some samples after 30 min machining by CDF without movement (**a**,**b**); with rotational movement (**c**,**d**); with rotation and oscillating movement (**e**,**f**).

**Figure 11 materials-15-06762-f011:**
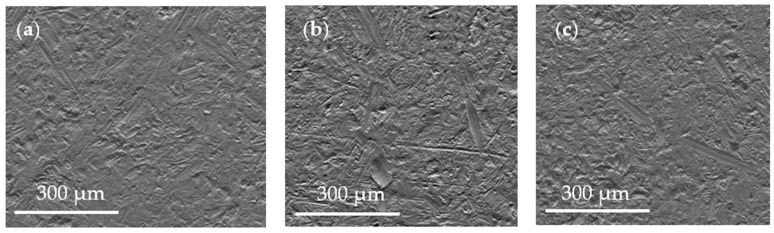
Images of the sample surface, obtained with SEM, after 45 min of smoothing: without movement (**a**); with rotational movement (**b**); with rotation and oscillating movement (**c**).

**Figure 12 materials-15-06762-f012:**
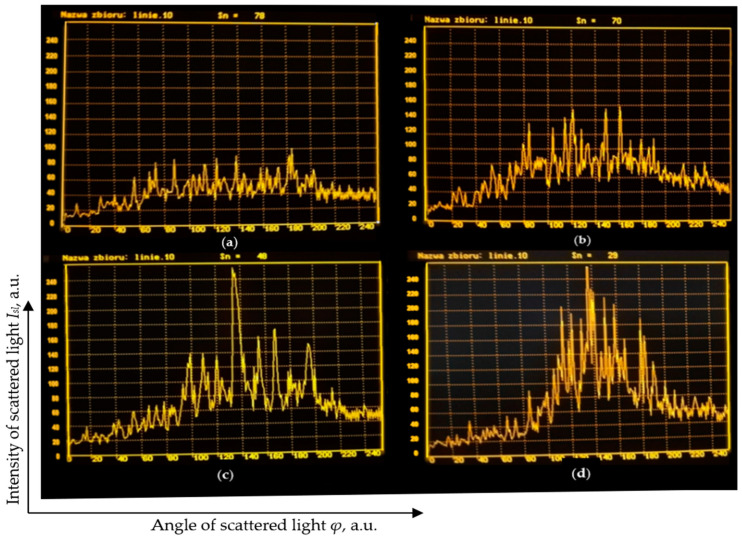
The angle distributions of the scattered light intensity obtained for sample without CDF machining (**a**); after 45 min of smoothing: without movement **(b**); with rotational movement (**c**); with rotation and oscillating movement (**d**).

**Figure 13 materials-15-06762-f013:**
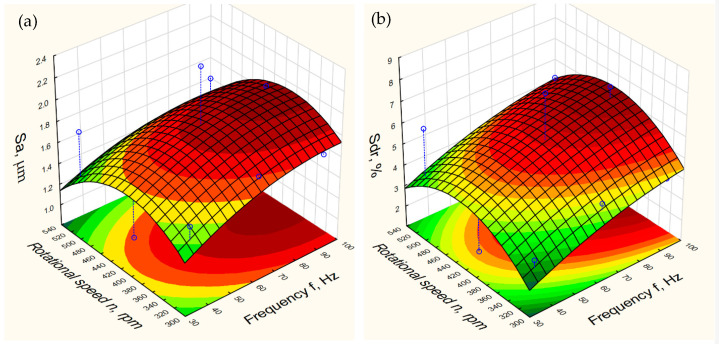
The three-dimensional plots of the arithmetic mean height *Sa* (**a**) and the variation of the developed interfacial area ratio *Sdr* (**b**) as functions of rotational speed *n* of the CDF machine disc and vibration frequency *f*.

**Table 1 materials-15-06762-t001:** Nominal chemical composition of the AISI 304 steel (in wt.%).

C	Cr	Ni	Si	Mn	P	S	N	Fe
≤0.07	17.0–19.5	8.0–10.5	≤1.0	≤2.0	≤0.045	≤0.045	≤0.011	Balance

**Table 2 materials-15-06762-t002:** Selected results obtained during the first stage of the experimental research.

Machining Time *t*, min	Surface Texture Parameters				
	*Sa*, µm	*Sdr*, %	*I*, %	*Ssc*, 1/µm	*Sdq*, µm/µm
	Machining the stationary workpiece				
15	2.24	8.06	61.5	0.075	0.412
30	2.22	6.93	57.5	0.073	0.380
45	1.96	6.05	55.5	0.067	0.355
	Machining with rotational movement of the workpiece				
15	2.31	8.82	60.4	0.084	0.432
30	1.98	5.30	57.1	0.061	0.331
45	1.67	3.72	44.4	0.057	0.276
	Machining with rotation and oscillation of the workpiece				
15	1.82	6.08	53.3	0.067	0.357
30	0.93	1.61	48.8	0.045	0.181
45	0.95	1.58	47.4	0.046	0.179

**Table 3 materials-15-06762-t003:** Selected results obtained during the second stage of experimental research.

Sample No.	*n* rpm	*f* Hz	Surface Texture Parameters					
			*Sa*, µm	*Sdr*, %	*I*, %	*Ssc*, 1/µm	*Sdq*, µm/µm	*SPc*, pks/mm^2^
1	312	35	1.55	3.23	40.1	0.053	0.257	1.07
2	312	65	1.71	4.36	49.9	0.061	0.302	1.00
3	312	95	1.63	4.06	40.2	0.055	0.290	1.13
4	416	35	1.07	1.72	42.1	0.046	0.187	0.25
5	416	65	2.38	7.76	49.9	0.075	0.402	4.20
6	416	95	1.94	6.76	49.2	0.072	0.377	6.52
7	520	35	1.72	5.84	57.1	0.068	0.350	2.38
8	520	65	0.97	2.25	31.8	0.051	0.226	0.38
9	520	95	1.70	5.62	49.2	0.066	0.344	3.95

## Data Availability

Not applicable.

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
