# Peer review of "Investigation of the Surface Treatment Process of AISI 304 Stainless Steel by Centrifugal Disc Finishing with the Use of an Active Workpiece Holder"

_materials, 2022, doi:10.3390/ma15196762_

Round 1

Reviewer 1 Report

Dear Authors,

I am not convinced with technical details of work and my concerns are listed below:

1.       What is the value of surface finish achieved using CDF process by other researchers. Mention the results in reported literature.

2.       Literature related to CDF process of year 2021 and 2022 is missing.

3.       What is the scope of this research work?

4.       Why the surface texture get improved 2nd stage. Explain with reason.

5.       What is the size of resin binders?

6.       Process parameter and range table is missing in manuscript.

7.       Clearly explain the CDF process mechanism.

8.       Combine the heading result and discussion.

9.       Need to rewrite the conclusion part and it should be precise. Results are not mentioned in conclusion part.

10.   Results are not highlighted in the abstract.

11. Reference style is not uniform and it should be as per journal requirement.

Author Response

Dear Editor,

Co-authors and I very much appreciated the reviewers' encouraging, critical and constructive comments on this manuscript. The comments have been very thorough and helpful in improving the manuscript. We strongly believe that the comments and suggestions have increased the scientific value of the revised manuscript in many places. We have taken them fully into account in the revision. The corrected or added parts of the manuscript are in the manuscript, highlighted in red.

We are submitting the corrected manuscript. The manuscript has been revised as per the comments given by the reviewers, and our responses to all the comments are as follows:

Reviewer #1:

  1. What is the value of surface finish achieved using CDF process by other researchers. Mention the results in reported literature.

Response: Thank you so much for your comments. The Introduction was supplemented with the results obtained by other researchers on the smoothing of the surface in the CDF process, presented in references.

The results of the research on the geometric structure of the surface after the CDF smoothing process are varied and depend on the used media, technological parameters and the movements of the workpieces holders. In the work Kopp M., E. Uhlmann: Potential of Robot-Guided Centrifugal Disc Finishing  [8], where the robot plays the role of a handle, the obtained roughness parameter of the smoothed steel surfaces of the smoothed surfaces is around Ra = 1 µm. In relation to the Sa parameter, this value may be even several times higher.

  1. Literature related to CDF process of year 2021 and 2022 is missing.

Response: Thanks for your question. We have completed the References with publications from the last two years, and we refer to them in the Introduction.

Publications from the period 2021 and 2022 regarding the CDF process have been included and citted in our article. They complement the current state of knowledge in this area.

List of added publications:

  1. Kopp, M.; Uhlmann, E. Potential of robot-guided centrifugal disc finishing. In: Production at the Leading Edge of Technology. Behrens, BA., Brosius, A., Drossel, WG., Hintze, W., Ihlenfeldt, S., Nyhuis, P. (eds) Lecture Notes in Production Engineering 2022, Springer, Cham. https://doi.org/10.1007/978-3-030-78424-9_32
  2. Uhlmann, E.; Kopp, M. Measurement and modeling of contact forces during robot-guided drag finishing. Procedia CIRP 2021, 102, 518–523. https://doi.org/10.1016/j.procir.2021.09.088

  1. What is the scope of this research work?

Response: Thanks for your question. In the additional section of the Introduction, we have expanded the explanation of the motivation, purpose, and scope of the research (lines 99-106).

We added in to the introduction:

The aim of the experimental research was to analyze the mechanism of shaping of the surface texture in the centrifugal disc finishing process, with the use of additional movements (vibration) of the worpieces mounted in the holder. The purpose of these researches are to assess the intensity and efficiency of this process. Such a scope of experimental research enables the analysis of the influence of the surface smoothing process in the adopted technical variants and allows to determine influence of the method of abrasive shapes on the shaping of the surface topography of the treated surface.

  1. Why the surface texture get improved 2nd stage. Explain with reason.

Response: We have taken reviewer’s comment in full consideration.

The research plan was divided into the following two stages:

  1. The research on the influence of the type of workpiece movement on the geometrical structure of its surface in relation to smoothing in a stationary holder.
  2. The research of the influence of the integrated rotation and vibration move of the workpiece on the shaping of the geometrical structure of its surface.

  1. What is the size of resin binders?

Response: We appreciate your question. Information about the shape and dimensions of the chips with a resin binder has been given on page 6, after their symbol.

The characteristic dimensions of the 02PP10 chips are 10 mm. (lines 160).

  1. Process parameter and range table is missing in manuscript.

Response: Thank you so much for your comment. (information is added in publication on lines 201-208).

After choosing the factors influencing on the research object, the ranges of the variability of the input factors were determined. Accepted ranges of the input factors for the research are:

- for the first stage of research:

1) machining time:       t=15,30 and 45, min,

- for the second stage of research:

2) rotational speed of the disc:  n=312,416 and 520, rpm

3) vibration frequency of the active holder:       f=35,65 and 95, Hz.

  1. Clearly explain the CDF process mechanism.

Response: We appreciate your comment. In the Introduction, the mechanism of the CDF process is further explained in more detail.

The smoothing process in rotary-cascade smoothing machines is based on the interaction of abrasive or polishing pieces placed in the working chamber with the surfaces of the workpieces, most often with a solution of supporting fluid, during forced rotation (Fig. 1).

In the smoothing process, there are a number of simultaneous elementary phenomena, which include:

  • micro-smoothing with abrasive grains from lumps of abrasive shaper,
  • mechanical abrasion of the unevenness of the smoothed surfaces due to the unevenness on the surface of the shapers,
  • plastic ridging by the blades of the shapers with a small caving,
  • grind in of the surface of the workpieces by abrasive grains and micro-grains, which are products of wear,
  • smoothing surface unevenness by plastic deformation,
  • chemical reactions and anodic processes occurring on the surfaces of workpieces in the presence of active chemical solutions and abrasive or polishing pieces.

These types of processes are classified, as a micro-smoothing processes with very small of the cutting depths of the order 10-8 – 10-5 of equivalent chip thickness heq and at the same time, high unit energy above 1000 J / mm3 (Fig. 2).

Value of specific Energy in the chamber of the finishing machine

Depending on the interaction between the contact surface with a specific instantaneous orientation, value of the contact force and velocity, different interaction mechanisms will be induced, i.e. plastic deformation, micro-cutting, grooving or scratching. In the case of the normal direction of action of the shapers on the treated surface, large local plastic deformations can be observed, which are result of the formation of the microcraters. On the other hand, the tangential movement of the shapers in relation to this surface, let for generating of the micro-cutting and scratching processes, because the speed of this movement is relatively high compared to the speed in the normal direction. In the intermediate states of the orientation of the treated surfaces, the cumulative effect of the abrasive blocks has a different character, which results in different character of the mechanisms of plastic deformation and micro-cutting and scratching.

For a more detailed discussion of the process mechanism, see publication [19, 24, 25].

  1. Combine the heading result and discussion.

Response: Thank you so much for your suggestion. The section Results and section Discussion have been merged.

  1. Explain the differences in the SEM images given in Figure 11.

Response: Thank you so much for your comment. (information is added in publication on lines 245-267).

Base on SEM images of the treated surfaces in such a research plan, it can be concluded that both the implementation of smoothing without additional workpieces movement and the active workpieces holder significantly influence on the process of shaping the geometric structure of the smoothed surfaces. This is confirmed by microscopic images of surfaces after 45 minutes of treatment in the accepted technological variants.

And so, in the case of smoothing without the movement of the object (Fig. 3a), some characteristic smoothing mechanisms can be distinguished on the shaper surface. Micro-cavities, which are the result of dynamic contacts of the tips of the abrasive blocks, have a dominant share here. On this basic structure, there are local traces of micro-cutting and grooving with few flashes as well as traces of scratching and reeling of the shapers on the smoothed surface. The dynamics of these interactions is relatively high and results from the long contact time of the abrasive shapers with the smoothed surface. In the case of smoothing with the rotation of the workpiece (Fig. 3b), the time of temporary contacts of the abrasive pieces is significantly shortened and a local change in the direction of their operation takes place as a result of the mutually complex movement. As a result, there is a local increase in the number of smoothing marks and plastic deformations. The smoothed surface is characterized by a greater degree of development and a relatively higher roughness. On the other hand, in the case of smoothing with rotational and vibrating motion (Fig. 3c), there is a cumulative interaction of these movements with the rotational (centrifugal) motion of the abrasive shapers. The time of their temporary contacts is shortened and the directions of its operation are changed, which leads to a relatively greater reduction in surface roughness.

  1. Need to rewrite the conclusion part and it should be precise. Findings are not mentioned in conclusion part.

Response: We appreciate your comment. The Conclusions section has been re-edited and highlights the most important research results. (information is added in publication on lines 334-352 and 361-365).

The results of the smoothing process in the CDF machine of AISI304 steel elements, mounted in an active holder with rotational and rotary-vibration movement and in a stationary holder, presented in the first stage of the research, indicate significant differences in the mechanism of shaping the geometric structure of the smoothed surfaces. These differences result from the diversified relative direction of movement of the abrasive shapers with the smoothed surface, the time of their contact and the number of accumulated contacts. It influences on the shaping of the smoothing traces originating from the elementary phenomena of micro-cutting, scratching, grooving, cratering, and the mechanism of shaping the surface structure. The best results in terms of shaping the geometric structure are achieved with the use of rotational and vibration movements of the workpiece. This method of processing also provides a significantly higher processing intensity compared to the processing carried out with the rotational of the workpiece itself, in particular with processing in a free state. It is a synergistic effect of the influence of a complex system of mutual movements of the object and working shapers, the multiplicity and dissimilarity of elementary phenomena occurring in the smoothing zone.

The method of shaping the geometric structure of the surface in the CDF machine with the use of an active workpiece holder, which gives the cumulative rotary-vibration motion, a lesser influence is observed by rotational speeds than by the vibration frequency of the workpiece.

  1. Results are not highlighted in the abstract.

Response: Thank you so much for your comment. The Abstract was supplemented with short information about the results obtained. (information is added in publication on lines 25-27).

The presented research results show that the use of an active holder, enabling rotation and oscillation of the workpiece, may lead to a more effective use of smoothing processes in the CDF machines.

  1. Reference style should be uniform and as per journal requirement.

Response: Thank you so much for your comment. The style of the references was corrected accordance with the requirements of the Materials journal.

Reviewer 2 Report

This kinematics can be found in a similar form in the Streamfinish system from Otec, which is used for the surface treatment of cutting tools such as drills. With regard to mass production, as described in the introduction, the modified system is no longer suitable, so that the motivation and the aim of the investigations should be revised with an adapted introduction. The significance of adapted kinematics in terms of machining efficiency is well analyzed and studied. The diagrams in Figure 13 require a more thorough interpretation. Which mechanisms lead to a reduction in roughness at both high and low speeds?

Author Response

Reviewer #2:

  1. This kinematics can be found in a similar form in the Streamfinish system from Otec, which is used for the surface treatment of cutting tools such as drills. With regard to mass production, as described in the introduction, the modified system is no longer suitable, so that the motivation and the aim of the investigations should be revised with an adapted introduction.

Response: Thank you so much for your valuable suggestion.

The kinematics of OTEC's Streamfinish smoothing process is different from that shown in this article. In the OTEC solution, the object in the holder only rotates. On the other hand, in our solution, in addition to the rotary motion, the object performs a vibrating motion, which results in a more complex system of generating the surface structure of the workpiece.

  1. The significance of adapted kinematics in terms of smoothing efficiency is well analyzed and studied.

Response: Thank you so much for your favorable comment.

The aim of the experimental research was to analyze the mechanism of shaping the surface texture in the process of the CDF machine, with the use of additional movements of the workpieces mounted in the holder. The purpose of these researches is to assess the intensity and efficiency of smoothing. Such a scope of experimental research enables the analysis of the course of the surface smoothing process in the adopted technical variants and allows to determine the influence of the method of influence of abrasive shapers on the shaping of the surface topography of the smoothed surface.

  1. The diagrams in Figure 13 require a more thorough interpretation. Which mechanisms lead to a reduction in roughness at both high and low speeds?

Response: Thank you so much for your comments.

The analysis of the graphs shows that the values of the Sdr and Sa parameters are more strongly dependent on the vibration frequency and increase with increasing frequency of it. This is undoubtedly the result of the concentration of smoothing marks on the smoothed surface. However, with regard to the rotational speed of the object, this relationship is non-monotonic, and its greatest influence occurs for its intermediate values. It follows that this activity does not have a significant impact on the generation of the number of smoothing marks and the degree of their concentration.

Reviewer 3 Report

The manuscript has several significant shortcomings that do not allow it to be published in its current form.

1.     The paper states that the main objectives of the study were to evaluate the mechanism of surface texture formation and study the efficiency of the machining process using an active workpiece holder, however, a detailed discussion of the results obtained is not disclosed in full.

2.     Thus, the text lacks specific information about the mechanisms of formation of the structure of the studied samples and does not provide mathematical justifications. In the work devoted to the study of the process of surface treatment by centrifugal processing, too much attention is paid to the description of the installation used. It is recommended to reconsider the relevance of Fig. 3, 4 and exclude images of the CAD model with a kinematic node, describing the principle of operation in the text only minimally.

3.     I think that not all scientifically significant articles have been analyzed. I believe that these articles can also be included in your review:  Konovalov, S., Ivanov, Y., Gromov, V., Panchenko, I. Fatigue-induced evolution of AISI 310S steel microstructure after electron beam treatment (2020) Materials, 13 (20), № 4567, pp. 1-13. DOI: 10.3390/ma13204567; Bai, X., Han, Y., Liaw, P.K., Wei, L. Effect of Ion Irradiation on Surface Microstructure and Nano-Hardness of SA508-IV Reactor Pressure Vessel Steel (2022) Journal of Materials Engineering and Performance, 31 (3), pp. 1981-1990.  DOI: 10.1007/s11665-021-06376-x; Dzyura, V., Maruschak, P. Optimizing the formation of hydraulic cylinder surfaces, taking into account their microrelief topography analyzed during different operations (2021) Machines, 9 (6), № 116. DOI: 10.3390/machines9060116

4.     Several controversial issues appear in the description of the materials used in the study. Instead of flat samples, it is optional to use samples with a complex shape (presence of holes, concave and convex areas, sharp corners) and evaluate the influence of geometric parameters on the efficiency of the machining process, which would correspond to the goals of evaluating the mechanism of surface texture formation depending on the intensity of processing . It should be noted that the paper does not describe in detail the properties of AISI 304 steel (both before and after machining of flat samples). The chemical composition of the metal without the properties of the material does not carry the necessary information, especially without comparison before and after processing.

5.     The work should reflect the measurement errors of the profilometer and the experimental data obtained (Table 2.3 and Fig. 6-8). Data on surface roughness obtained by the light scattering method (Fig. 13) are considered redundant due to the presented surface maps obtained on the profilometer (Fig. 6, 7) as well as the error, which is extremely difficult to evaluate in this method. Instead of maps constructed by the light scattering method, it is recommended to add an image of the surface in transverse microsections obtained on a light or scanning microscope with overlaying information about the height of the peaks, thereby comparing the results with the profilometer data and additionally determining the error of the device.

6.     Chapter 4 indicates the significant influence of the oscillation frequency of an electromagnetic vibration installation, however, in a series of experiments, the surface treatment of the samples under study using cyclic oscillations was carried out exclusively together with rotation. The parameter was not taken into account in the work, it is recommended to conduct a series of additional experiments. Additionally, it is proposed to compare alternative methods of finishing according to the criteria of complexity and time of processing, economic feasibility.

Author Response

Reviewer #3:

  1. The paper states that the main objectives of the study were to evaluate the mechanism of surface texture formation and study the efficiency of the smoothing process using an active workpiece holder, however, a detailed discussion of the results obtained is not disclosed in full.

Response:

Thank you so much for your comments. The aim of the experimental research was to analyze the mechanism of shaping the surface texture in the process of CDF, with the use of additional movements of the workpieces mounted in the holder. The purpose of these researches is to assess the intensity and efficiency of smoothing. Such a scope of experimental research enables the analysis of the course of the surface smoothing process in the adopted technical variants and allows to determine the influence of the method of influence of abrasive shapers on the shaping of the surface topography of the smoothed surface.

The research plan was divided into the following two stages:

  1. The research on the influence of the type of workpiece movement on the geometrical structure of its surface in relation to smoothing in a stationary holder.
  2. The research of the influence of the integrated rotation and vibration move of the workpiece on the shaping of the geometrical structure of its surface.

  1. Thus, the text lacks specific information about the mechanisms of formation of the structure of the studied samples and does not provide mathematical justifications. In the work devoted to the study of the process of surface treatment by centrifugal processing, too much attention is paid to the description of the installation used. It is recommended to reconsider the relevance of Fig. 3, 4 and exclude images of the CAD model with a kinematic node, describing the principle of operation in the text only minimally.

Response: Thank you so much for your comment.

Base on SEM images of the treated surfaces in such a research plan, it can be concluded that both the implementation of smoothing without additional workpieces movement and the active workpieces holder significantly influence on the process of shaping the geomet-ric structure of the smoothed surfaces. This is confirmed by microscopic images of surfac-es after 45 minutes of treatment in the accepted technological variants.

And so, in the case of smoothing without the movement of the object (Fig. 11a), some characteristic smoothing mechanisms can be distinguished on the shaper surface. Micro-cavities, which are the result of dynamic contacts of the tips of the abrasive blocks, have a dominant share here. On this basic structure, there are local traces of micro-cutting and grooving with few flashes as well as traces of scratching and reeling of the shapers on the smoothed surface. The dynamics of these interactions is relatively high and results from the long contact time of the abrasive shapers with the smoothed surface. In the case of smoothing with the rotation of the workpiece (Fig. 11b), the time of temporary contacts of the abrasive pieces is significantly shortened and a local change in the direction of their operation takes place as a result of the mutually complex movement. As a result, there is a local increase in the number of smoothing marks and plastic deformations. The smoothed surface is characterized by a greater degree of development and a relatively higher rough-ness. On the other hand, in the case of smoothing with rotational and vibrating motion (Fig. 11c), there is a cumulative interaction of these movements with the rotational (centrifugal) motion of the abrasive shapers. The time of their temporary contacts is shortened and the directions of its operation are changed, which leads to a relatively greater reduction in surface roughness.

  1. I think that not all scientifically significant articles have been analyzed. I believe that these articles can also be included in your review:  Konovalov, S., Ivanov, Y., Gromov, V., Panchenko, I. Fatigue-induced evolution of AISI 310S steel microstructure after electron beam treatment (2020) Materials, 13 (20), № 4567, pp. 1-13. DOI: 10.3390/ma13204567; Bai, X., Han, Y., Liaw, P.K., Wei, L. Effect of Ion Irradiation on Surface Microstructure and Nano-Hardness of SA508-IV Reactor Pressure Vessel Steel (2022) Journal of Materials Engineering and Performance, 31 (3), pp. 1981-1990.  DOI: 10.1007/s11665-021-06376-x; Dzyura, V., Maruschak, P. Optimizing the formation of hydraulic cylinder surfaces, taking into account their microrelief topography analyzed during different operations (2021) Machines, 9 (6), № 116. DOI: 10.3390/machines9060116

Response:

Thank you so much for your valuable suggestion. The publications proposed by the Reviewer are beyond the scope of the article, therefore they were not included.

Publications from the period 2021 and 2022 regarding the CDF process have been included and citted in our article. They complement the current state of knowledge in this area.

List of added publications:

  1. Kopp, M.; Uhlmann, E. Potential of robot-guided centrifugal disc finishing. In: Production at the Leading Edge of Technology. Behrens, BA., Brosius, A., Drossel, WG., Hintze, W., Ihlenfeldt, S., Nyhuis, P. (eds) Lecture Notes in Production Engineering 2022, Springer, Cham. https://doi.org/10.1007/978-3-030-78424-9_32
  2. Uhlmann, E.; Kopp, M. Measurement and modeling of contact forces during robot-guided drag finishing. Procedia CIRP 2021, 102, 518–523. https://doi.org/10.1016/j.procir.2021.09.088
  3. Several controversial issues appear in the description of the materials used in the study. Instead of flat samples, it is optional to use samples with a complex shape (presence of holes, concave and convex areas, sharp corners) and evaluate the influence of geometric parameters on the efficiency of the smoothing process, which would correspond to the goals of evaluating the mechanism of surface texture formation depending on the intensity of processing . It should be noted that the paper does not describe in detail the properties of AISI 304 steel (both before and after smoothing of flat samples). The chemical composition of the metal without the properties of the material does not carry the necessary information, especially without comparison before and after processing.

Response:

The possibilities of processing samples with more complex shapes and with internal holes, indicated by the Reviewer, are planned in the next stages of the research program. We are also working on shaping locally different surface structures by masking unprocessed parts of the surface. We also work on the properties of smoothed surfaces taking into account technological heritage, determining how the effects of previous operations affect the quality and efficiency of finishing.

  1. The work should reflect the measurement errors of the profilometer and the experimental data obtained (Table 2.3 and Fig. 6-8). Data on surface roughness obtained by the light scattering method (Fig. 13) are considered redundant due to the presented surface maps obtained on the profilometer (Fig. 6, 7) as well as the error, which is extremely difficult to evaluate in this method. Instead of maps constructed by the light scattering method, it is recommended to add an image of the surface in transverse microsections obtained on a light or scanning microscope with overlaying information about the height of the peaks, thereby comparing the results with the profilometer data and additionally determining the error of the device.

Response:

The scattering distributions of the light reflected from the processed surfaces were used to assess its sensitivity to distinguishing the surface structure formed during the processing of the workpiece without moving, with rotational motion and with rotation-vibration motion. It will be used in subsequent research programs.

Information about Multisensory optical profilometer CLI2000 Taylor-Hobson (Leicester, Great Britain)

Components: laser triangulation sensor LK-031 (Keyence Corp., Osaka, Japan)

Features (sensor): scanning frequency: 2000 Hz, measuring range: 10 mm, resolution: 1 μm (vertical), 30 µm (lateral), measuring slope: 40°, speed: 30 mm/s

Features (instrument): measuring capacity: 200× 200×200 mm, axis traverse length: 200 mm, axis resolution: 0.5 μm, dimensions: 800×800×800 mm, measuring speed: 0.5, 1, 5, 10, 15 and 30 mm/s, positioning speed: 30 mm/s

Software: Talyscan CLI 2000 2.6.1+ TalyMap Silver 4.1.2 (Digital Surf, Besançon, France)

An area with dimensions (x, y axis): 4.8×4.8 mm was measured on each samples. The number of the profiles (y axis) was 321. The distance between the profiles (y axis) was 15 μm. The number of the profile points (x axis) was 2401. The distance between the profile points (x axis) was 2 μm. The measuring time was 4020 s.

  1. Chapter 4 indicates the significant influence of the oscillation frequency of an electromagnetic vibration installation, however, in a series of experiments, the surface treatment of the samples under study using cyclic oscillations was carried out exclusively together with rotation. The parameter was not taken into account in the work, it is recommended to conduct a series of additional experiments. Additionally, it is proposed to compare alternative methods of finishing according to the criteria of complexity and time of processing, economic feasibility.

Response:

As already mentioned in answer 1, the research program was divided into the following two stages:

  1. The research on the influence of the type of workpiece movement on the geometrical structure of its surface in relation to smoothing in a stationary holder.
  2. The research of the influence of the integrated rotation and vibration move of the workpiece on the shaping of the geometrical structure of its surface.

In the first stage, the influence of smoothing without the workpiece movement, with rotational movement and rotation-vibration movement was analyzed. This third form of exercise has been shown to provide the best results. Therefore, in the second stage, the influence of the parameters of these movements on the smoothing efficiency was investigated. As suggested by the Reviewer, we will try to compare alternative methods of surface finishing with the use of other solutions for object grips.

Round 2

Reviewer 1 Report

1Dear Authors,

I am not convinced with technical details of work and my concerns are listed below:

11. Selected process Parameters with their range should be mentioned in the table.

22. Conclusion section should be precised and to the point. Mention the value of surface roughness achieved by CFD process for both stages in the conclusion.

33. Reference style should be uniform and as per journal standards.

44. Need to rewrite the abstract part and also highlight the results of surface roughness

55. Add the CFD process mechanism in the manuscript.

Author Response

Reviewer #1:

  1. Selected process Parameters with their range should be mentioned in the table.

Response: Thank you so much for your comment. We added information about the range parameters for each stage of researches in text (in lines 233-242) to previous versions of article. The value for each point of research is presented in Table 2 and Table 3.

  1. Conclusion section should be precised and to the point. Mention the value of surface roughness achieved by CFD process for both stages in the conclusion.

Response: Thank you so much for your comment. The dependence of the surface roughness parameters is given in Table 2, depending on the processing time. The dependence of the surface roughness parameters is given in Table 3, depending on the processing parameters and the kinematics of the process.

  1. Reference style should be uniform and as per journal standards.

Response: Thank you so much for your comment. The style of the references was corrected accordance with the requirements of the Materials journal.

  1. Need to rewrite the abstract part and also highlight the results of surface roughness

Response: Thank you so much for your comments. The Introduction was supplemented with the results of surface roughness. We add information about surface roughness into the introduction (see line 27-33)

The analysis of the results shows that the values of the Sdr and Sa parameters are more strongly dependent on the vibration frequency and increase with increasing frequency of it. This is undoubtedly the result of the concentration of smoothing marks on the smoothed surface. However, with regard to the rotational speed of the object, this relationship is non-monotonic, and its greatest influence occurs for its intermediate values. It follows that this activity does not have a significant impact on the generation of the number of smoothing marks and the degree of their concentration.

  1. Add the CFD process mechanism in the manuscript.

Response: Thank you so much for your comments. As reviewer recommend we added information about CFD process in text (lines 42-67)

Reviewer 3 Report

Authors have made corrections. No new comments.

Author Response

Reviewer accept our changes in previous version. Thank you very much.